# Effects of Maternal Protein Supplementation at Mid-Gestation of Cows on Intake, Digestibility, and Feeding Behavior of the Offspring

**DOI:** 10.3390/ani12202865

**Published:** 2022-10-20

**Authors:** Karolina Batista Nascimento, Matheus Castilho Galvão, Javier Andrés Moreno Meneses, Gabriel Miranda Moreira, German Darío Ramírez-Zamudio, Stefania Priscilla de Souza, Ligia Dias Prezotto, Luthesco Haddad Lima Chalfun, Marcio de Souza Duarte, Daniel Rume Casagrande, Mateus Pies Gionbelli

**Affiliations:** 1Department of Animal Science, Universidade Federal de Lavras, Lavras 37200-900, MG, Brazil; 2Department of Veterinary Medicine and Animal Science, Universidad de Ciencias Aplicadas y Ambientales, Cartagena 130001, Bolivar, Colombia; 3Timac Agro Brazil, Ribeirão Preto 14026-282, SP, Brazil; 4Department of Research Centers, Montana State University, Bozeman, MT 59501, USA; 5Department of Veterinary Medicine, Centro Universitário de Lavras, Lavras 37203-593, MG, Brazil; 6Department of Animal Bioscience, University of Guelph, Guelph, ON N1G 2W1, Canada

**Keywords:** appetite, fetal programming, organogenesis, sexual dimorphism, zebu beef cows

## Abstract

**Simple Summary:**

It is well known that intrauterine growth-restricted offspring present physiologic and metabolic modifications later in life. Therefore, understanding the impact of the maternal nutritional plane on feed intake patterns may lead to new feeding strategies to improve the feed efficiency and performance of beef cattle. In this study, we have evaluated the effect of maternal protein supplementation (PS) during mid-gestation and its interaction with the offspring’s sex on the voluntary feed intake and nutrient use of the progeny. Our results show that PS during mid-gestation increases performance and improves the voluntary feed intake of the offspring. Prenatal PS induced favorable changes in the feeding behavior, in which CON offspring spent more time per day eating supplements and ruminating but spent fewer periods in idleness. Maternal protein restriction demonstrated positive effects on the ability to digest the nutrients of male calves. In summary, our data show that PS alters the gain potential, indirectly demonstrating an increase in their nutritional requirements. In contrast, protein restriction causes a compensatory mechanism on the offspring’s nutrient useability in a sex-dependent manner, increasing the total tract digestibility of nutrients in response to a growth-restriction exposure during intrauterine development.

**Abstract:**

This study aimed to assess the effects of maternal protein supplementation and offspring sex (OS) on the intake parameters of the offspring. Forty-three Tabapuã cows were randomly allocated in the following treatments: protein supplementation (PS) during days 100–200 of gestation (RES, 5.5% total crude protein (CP), *n* = 2, or CON, 10% total CP, *n* = 19) and OS (females, *n* = 20; males, *n* = 23). The offspring were evaluated during the cow–calf (0–210 days), backgrounding (255–320 days), growing 1 (321–381 days), and growing 2 (382–445 days) phases. The CON offspring tended to present higher dry matter intake (DMI) at weaning (*p* = 0.06). The CON males presented lower digestibility of major diet components in the growing 2 phase (*p* ≤ 0.02). The CON offspring spent 52% more time per day eating supplements at 100 days and 17% less time in idleness at 210 days. The CON males spent 15 min more per day ruminating than RES males in the feedlot phase (*p* = 0.01). We concluded that protein supplementation over gestation alters the offspring feed intake pattern as a whole, while protein restriction promotes compensatory responses on nutrient digestibility in males.

## 1. Introduction

Meat production is an essential part of the world economy, with substantive contributions to local, national, and international trade [1]. There may be multiple paths to the future of meat production, but challenges to how it is produced will be under increasing pressure in the immediate and the foreseeable future [2]. In tropical areas, the vast majority of beef is produced under grazing conditions [3]. Seasonality affects forage production and its nutritional value , and the protein content of the forage is the main limitation related to animal feed intake [4]. To overcome a possible low nutritional value of the forage in tropical grazing systems, the breeding and calving seasons are usually programmed to overlap the rainy period [5] as a strategy to attend to the greater nutrient requirements for lactation. However, pregnant cows raised under extensive systems experience this extremely unfavorable nutritional scenario during mid-to-late gestation [6]. Thus, the use of nutritional corrections for pregnant cows exposed to the productive seasonality of pastures (such as protein supplementation) may be an interesting alternative to reduce the deleterious effects on the cows’ longevity and on their offspring [7,8].

It has been well documented that alterations in the prenatal environment affect the productivity of the offspring later in life [9,10,11,12]. Moreover, the maternal nutrition plans during pregnancy affect not only the offspring’s body weight [12], but also their post-natal growth rate, growth efficiency, and body gain composition [13,14,15], which in turn, may lead to changes in their nutritional requirements [13]. Consistently, available evidence from Greenwood et al. [15] demonstrated that low-birth-weight lambs had 30% lower energy requirements than their high-birth-weight contemporaries. Thus, once the nutritional requirements trigger the animals’ feed intake [16], this available evidence indirectly suggests that the feeding intake pattern later in life may be shaped by the nutritional conditions during intrauterine development. It has also been shown that maternal nutrition changes the dynamic of the hypothalamic circuit, which controls the feed intake [17,18,19], which in turn alters the offspring’s voluntary feed intake [17,20,21]. Therefore, based on the existing data, we hypothesized that exposure to nutritional insults during pregnancy could lead to changes in the offspring’s feed intake in a global manner, affecting the nutritional management decisions of beef cattle producers.

There is also available evidence that the mass of several organs [22,23], mRNA expression of membrane transporters in the small intestine (the main site of nutrient absorption) [24], digestive enzyme activity [25], and energy metabolic pathways in the liver [26] of the offspring may be shaped by nutritional conditions over gestation. Thus, based on the aforementioned, it is also possible that nutritional disturbances during intrauterine development affect the offspring’s nutrient digestibility in the long term. Nevertheless, little is known on if maternal nutrition during gestation in beef cattle may affect the performance of the offspring through changes in their nutrient use capacity. Therefore, this study also aimed to explore this knowledge gap, considering that nutritional perturbations during gestation may promote compensatory responses on the nutrient digestibility of the offspring.

Moreover, some studies on farm animals [5,7,27,28] support that the fetal programming responses occur in a sex-dependent manner. This sexual dimorphism may be triggered by the different DNA methylation patterns between male and female embryos over the pregnancy course, influencing epigenetic responses between the genders [29], as well as by the different steroid exposure during intrauterine development [30]. The maternal tissue mobilization intensity also differs between cows carrying female or male fetuses. As an adaptive mechanism [31], under a nutritionally challenging scenario, dams seem to invest more in female than male fetuses [32], making the fetal programming responses sex-dependent [33].

In summary, this study aimed to evaluate the effects of protein supplementation during mid-gestation in zebu beef cows and the offspring’s sex-dependent interaction on intake parameters, behavior, digestibility, feed efficiency, and liver size in the offspring.

## 2. Materials and Methods

This study was performed in the Beef Cattle Facility at the Universidade Federal de Lavras (UFLA—Lavras, Minas Gerais, Brazil) and was performed in two stages of two and half years each, with the same experimental procedures. Each period included the gestation of beef cows and the offspring evaluation from birth to 445 days of age. Animal welfare and all procedures were previously approved by the Brazilian Ethics Committee on Animal Use of UFLA (Protocol No. 015/17).

### 2.1. Animals, Housing, and Feeding

Forty-three purebred Tabapuã (*Bos taurus indicus*) multiparous cows (490.5 ± 17.8 kg of initial body weight (BW)) were artificially inseminated using semen from 3 different bulls. Fetal sex was detected at 60 days of gestation using ultrasound scans by a trained professional, characterizing 20 female and 23 male embryos. Cows were individually allotted in pens during mid-gestation, and at 102 ± 5 days of gestation, they were randomly divided into two groups: restricted (RES), fed basal diet (corn silage + sugarcane bagasse, achieving 5.5% CP plus a mineral mixture) (*n* = 24), or supplemented (CON), fed basal diet plus protein supplementation (40% CP at the level of 3.5 g/kg of BW, *n* = 19). The experimental diet (DM basis) provided to the cows from 102 ± 5 to 208 ± 6 days of gestation was based on 75% corn silage, 25% sugar cane bagasse, and a macro and micro mineral supplement provided ad libitum. For cows in the CON group, the protein supplement consists of a 50:50 mixture of soybean meal and a commercial supplement (Probeef Proteinado Sprint, Cargill Nutrição Animal, Itapira, SP, Brazil). The nutrient composition (per kilogram) from the commercial supplement was: 70 g Ca (max), 50 g Ca (min), 15 mg Co (min), 255 mg Cu (min), 15 g S (min), 2000 mg F (max), 20 g P (min), 15 mg I (min), 510 mg Mn (min), 340 NPN protein eq. (max), 450 g CP (min), 4 mg Se (min), 95 g Na (min), 850 mg Zn (min), and 50 mg Flavomycin. In both periods of gestation, animals were fed twice a day (at 7:00 a.m. and 1:00 p.m.). The average chemical composition of the feedstuffs used during the experimental period is described in Table 1.

From 208 ± 6 days of gestation until the parturition, all cows were fed ad libitum with corn silage and mineral mixture ad libitum. During the gestational period, the feed intake was measured as the difference between the amounts of ingredients supplied and the orts obtained. Feeds were sampled daily, and weekly composite samples were obtained for further analysis. Refusals were collected daily and sampled weekly for further analysis.

After parturition, cow–calf pairs were allocated in a *Brachiaria brizantha* cv. Marandu pasture area (y1 (in a DM basis): DM = 29.6%, CP = 14.5%, neutral detergent fiber (NDF) = 66.3%; y2: DM = 29.8%, CP = 13.9%, NDF = 75.7%) and raised in an intensive grazing management. The animals were managed under a continuous stocking method in a single pasture area (70,000 m^2^) with no subdivisions. Thus, the pasture area was the same for all cows and their calves during the entire cow–calf phase. Moreover, in the second repetition of the experiment (y2), the same pasture area used in the first experimental year (y1) was used for matrices and their offspring allocation. Cows received mineral supplementation, and calves received a commercial supplement (Probeef maxima creep, Cargill Nutrição Animal, Itapira, SP, Brazil) at the level of 5 to 7 g/kg of BW/day. Nutrient composition per kilogram of product was: 200 g crude protein (min), 20 g Ca (max), 30 g Ca (min), 3 mg Co (min), 51 mg Cu (min), 1 mg (min) Cr, 10.4 g dextrose, 3000 mg S (min), 0.42 ethoxyquin (min), 2000 mg F (max), 6000 mg P (min), 3 mg I (min), 700 mg mananas (min), 108 mg Mn (min), 60 mg monensin, 0.90 mg Se, 10 g Na (min), 12,000 UI vitamin A (min), 15,000 UI vitamin D3 (min), 50 UI vitamin E (min), 180 mg Zn (min).

Calves were weaned at 210 days of age and remained in a grazing system until 255 days of age, when they were transferred to a feedlot. Animals were confined for 190 days, and this period was divided into three phases with different diets, as follows: backgrounding (255–320 days of average age), growing 1 (321–381 days of average age), and growing 2 (382–445 days of average age) phases. Males and females were fed ad libitum and received diets with the same roughage:concentrate ratio during the backgrounding (72:28), growing 1 (65:35), and growing 2 phases (30:70). The diets provided during the feedlot period were offered as total mixed rations (TMRs). Nevertheless, the concentrate formulation for backgrounding and growing 1 phase was different for males and females, aiming to fully meet the specific nutritional requirements for each sex (Table 2). It is well known that males and females have different nutritional requirements, which is explained by the differences in growth dynamics between the sexes. Therefore, in the backgrounding and growing 1 phases (when the sex differences in growth trajectory were accentuated), males and females were fed with the same roughage:concentrate proportion, but the concentrate formulation was different between the sexes (to fully meet the specific requirements for each sexual category; Table 2). Thus, if we did not use diets compatible with the nutritional and energy needs of males and females separately, we could mask our responses. On the other hand, in the growing 2 phase, we considered that both sexes had reached the ”plateau” of the growth trajectory, which in turn dispensed these adjustments. In this sense, in the growing 2 phase, males and females were fed the same diets (Table 2).

Samples of corn silage were collected three times per week; ort samples were collected daily, and concentrated representative samples were collected after every ration mixture confection in the feed factory. All samples were stored at −20°C until analyses. Heifers and steers had free access to water and were fed twice daily (at 7:00 a.m. and 1:00 p.m.). 

From the 43 progeny used in this experiment, 2 were born extremely weak and died before 7 days of age, both being RES males. Moreover, after weaning, one female from the RES group died due to a factor unrelated to the experiment. Thus, the data presented from the cow–calf phase are from results obtained from 41 animals, while data from weaning to the end of the experimental period are from 40 animals.

### 2.2. Measurements

#### 2.2.1. Body Weight Measurements

The maternal body weight measurements were performed at 100 (beginning of treatment application), 200 (end of supplementation period), and 270 (pre-parturition) days of gestation, in the morning and after 16 h fasting. During the same gestational periods, the body condition scores (BCS) were taken by three trained evaluators using a scale from 1 to 9 points (in which 1 = emaciated and 9 = obese cow). The final score was defined as the average values obtained from each evaluator. In the cow–calf phase, the body weight measurements were taken with calves in fasting. Before the BW determination, calves were isolated from their dams for 12 h. At the feedlot, the animals were weighed before the morning feeding.

#### 2.2.2. Intake, Digestibility Trials, and Feed Efficiency

In the cow–calf phase, calves were subject to two digestibility trials at 120 and 210 days of age. For estimation of forage and supplement intake and fecal production quantification, the indigestible neutral detergent fiber (NDFi) [34,35,36,37], chromium oxide (CrO_2_) [37,38], and titanium dioxide (TiO_2_) [37,39,40] were used as indicators. The titanium dioxide was provided one time per day in the morning (0700 h) wrapped in paper cartridges in doses of 10 g of TiO_2_ per animal administered using an esophagus probe. The chromium oxide was provided mixed in the calves’ supplement at a concentration of 0.5% of the supplement consumed during the digestibility trials. Both indicators were provided for 10 consecutive days, and the fecal samples were collected by spot technique in the morning (7:00 a.m.) and afternoon (6:30 p.m.) of the last four days of indicator supply. Pasture samples were obtained on days 6, 7, 8, and 9 of the digestibility period through the manual grazing simulation technique. For pasture sample collection, the total area (70,000 m^2^) was subdivided into 5 parts to ensure a representative sample collection.

The fecal production (FP), the supplement dry matter intake (DMI*_sup_*), and the forage dry matter intake (DMI*_forage_*) were estimated according to Equations (1)–(3), respectively:(1)$$FP\left(\mathrm{kg}/\mathrm{day}\right)=\frac{I_{supplied}}{I_{feces}}$$
where *FP* = fecal production, *I_supplied_* = concentration of indicator supplied to the animal (kg/day), and *I_feces_* = concentration of indicator in the feces (kg/kg).
(2)$$DMI_{sup}\left(\mathrm{kg}/\mathrm{day}\right)=\frac{\left(FP\times I_{feces}\right)}{I_{sup}}$$
where *FP* = fecal production (kg/day), *If_eces_* = indicator concentration in the feces (kg/kg), and *I_sup_* = indicator concentration in the supplement (kg/kg).
(3)$$DMI_{forage}\left(\mathrm{kg}/\mathrm{day}\right)=\frac{FP\times I_{feces}}{NDFi_{forage}}$$
where *FP* = fecal production (kg/day), *NDFi_feces_* = concentration of *NDFi* in the feces (kg/kg), and *NDFi_forage_* = concentration of *NDF* in roughage (kg/kg).

The milk intake was calculated as the product between the daily milk production and milk DM content. During each offspring digestibility trial performed, the cows were hand-milked once to determine milk intake by calves, and calves were isolated from their dams for approximately 12 h before each procedure. Milk letdown was stimulated by injection of 2 mL of oxytocin (Ocitocina Forte UCB, Uzinas Chimicas Brasileiras S/A, Jaboticabal, Brazil) in the morning (0600 h). Then, the milk was weighed, and milk samples were collected in sterile vials containing a bronopol tablet (D & F Control Systems Inc., San Ramon, CA, USA). Tubes were kept at 4 °C until analysis. Daily milk yield (MY) was calculated as follows:(4)$$MY\left(\mathrm{kg}/\mathrm{day}\right)=\frac{MY_{morning}\left(\mathrm{kg}/\mathrm{day}\right)}{\left(Time_{1}+1\right)-Time_{2}}$$
where *MY* = milk yield, *Time*_1_ = hour of milk procedure ending, and *Time*_2_ = hour of calf isolation of dams [41].

In the post-weaning phase, there was one digestibility trial per phase during the feedlot period. Trials were performed at 310, 370, and 425 days of age in the backgrounding, growing 1, and growing 2 phases, respectively. Fecal samples were collected by hand grab technique directly from the rectum, for 5 consecutive days at different times at each day (day 1 = 0600 h, day 2 = 0900 h, day 3 = 1200 h, day 4 = 1500 h, and day 5 = 1800 h). Fecal sampling was performed over different daily periods to ensure a representative composite sample for each animal throughout the daytime during the digestibility trial. The orts were recorded daily before the morning feeding, and DMI was estimated for each animal. Throughout the trials, daily samples of corn silage and orts were collected, along with representative samples of concentrate used in each phase. Samples were stored at −20 °C until analyses. The NDFi was used as an indicator to measure fecal production, which was estimated using Equation (1).

During cow–calf and post-weaning phases, the apparent total tract digestibility of DM, organic matter (OM), CP, and NDF expressed in g/kg of nutrients was determined by the difference between intake and the content in feces divided by intake. Feed efficiency for gain was obtained as the ratio between the average daily gain (ADG) and the DMI.

#### 2.2.3. Chemical Analyses

All samples (feed, ort, and fecal samples) were individually dried in a forced dry oven (65 °C) for 72 h and ground (Wiley mill; A. H. Thomas, Philadelphia, PA, USA) in 1 and 2 mm bolters. Samples were chemically analyzed following AOAC [42] methods (CP, 984.13; ash, 119 942.05; EE, 920.39; moisture, 934.01). The neutral detergent fiber (NDF) content was analyzed according to Van Soest et al. [43] using heat-stable α-amylase. Non-fibrous carbohydrates (NFCs) were calculated according to Detmann and Valadares Filho [44]. Total digestible nutrients (TDNs) were calculated as: TDN = % digestible CP + % digestible NDF + % NFC digestible + 2.25% digestible EE [45]. The quantification of Cr_2_O and TiO_2_ was performed using atomic absorption spectrophotometry and colorimetric determination, according to Kimura et al. [38] and Myers et al. [39], respectively. The NDFi quantification was performed according to Valente et al. [34], via in situ incubation. Milk samples were analyzed for composition determination in a commercial laboratory using an infrared analyzer (Bentley2000, Bently Instruments, Chasca, MN, USA).

#### 2.2.4. Feeding Behavior

The feeding behavior trials were performed at 100 and 210 days of age in the cow–calf phase and in the middle of the feedlot period (360 days of age) in the growing 1 phase. In each trial, the feeding behavior was monitored for 48 h uninterrupted (day and night on two consecutive days) by human observation (at least three observers per hour). To judge the activities performed by each animal, each observer was assigned to a group of animals (~1/3 of the animals per observer) for evaluation. During the trials, each observer was positioned in a place where they could observe the animal without affecting its natural behavior. During the night periods, the installations’ artificial lights were kept off, and the feeding behavior evaluations were performed with the aid of flashlights. Calves were monitored for frequency and time of milk and supplement intake. Grazing, rumination, idleness, or other activities (locomotion or water intake) were evaluated every 10 min in the weaning phase. At 360 days of age, steers and heifers were monitored for the time of rumination, voluntary intake, idleness, or other activities (locomotion or water intake), also at intervals of 10 min. Feeding behavior measurements were converted to continuous time through the product between the frequency of the respective activity (during 48 h) and the time interval of 10 min. Subsequently, we obtained the time spent on each activity expressed in minutes per day, dividing the value obtained by 2880 (which represents the total minutes within two days).

### 2.3. Statistical Analysis

All data analyses were performed using SAS 9.2 (Statistical Analysis System Institute, Inc., Cary, NC, USA). In the statistical analysis of the cow–calf phase and growing 2 data, the maternal dietary treatment (2 levels) and offspring sex (2 levels) were considered fixed effects. Due to the dietetic differences for males and females in the backgrounding and growing 1 phases, only the maternal dietary treatment was considered a fixed effect in the statistical model for this data analysis. Therefore, in these periods, the data were analyzed separately for males and females (i.e., RES male vs. CON male, RES female vs. CON female). The period in which the experiment was performed (year 1 and year 2) and the parents’ index of genetic merit expected for growth traits (GEN) were considered random effects. The GEN was calculated using the data available on the Tabapuã Genetical Enhancement Program using the information on expected progeny difference (EPD) from the animal parents. The EPD parents’ data consisted of weight at weaning, weight at 12 months, and weight at 18 months. When pertinent (*p* < 0.05), the dam’s numbers of parturitions, empty weight at 100 days of gestation, cow size, BSC with 100 days of pregnancy, and gestation length and the offspring age at the evaluation period and body weight at the evaluation period were used as covariates. When not pertinent, they were taken from the model.

Data from the cow–calf and growing 2 phases were analyzed using the following model:*Yijkl* = *μ* + *Di* + *Sj* + (*DS*)*ij* + *Tk* + *Gl* + *εijkl*

where *Yijk* is the observed measurement; *μ* is the overall mean; *Di* is the fixed effect of the *ith* level of maternal dietary treatment; *Sj* is the fixed effect of the *jth* level of offspring sex; *DSij* is the interaction between *D* and *S*; *Tk* is the random effect of the kth period (year of the experiment); *Gl* is the random effect of the lth index of dam’s genetic merit expected for growth traits; and *εijkl* is the random error associated with *Yijkl*, with *eijkl ~N* (*0,**σe2*).

Data from the backgrounding and growing 1 phases were analyzed using the following model:*Yijk* = *μ* + *Di* + *Tj* + *Gk* + *εijk*

where *Yijk* is the observed measurement; *μ* is the overall mean; *Di* is the fixed effect of the *ith* level of maternal dietary treatment; *Tj* is the random effect of the *jth* period (year of the experiment); *Gk* is the random effect of the *kth* index of dam’s genetic merit expected for growth traits; and *εijk* is the random error associated with *Yijk*, with *eijk ~N* (*0,**σe2*).

Before the final analyses, studentized residuals were removed when not within ±3 standard deviations, and normality (*p* > 0.05) was assessed using the Shapiro–Wilk test. Least-square means were separated using Fisher’s least significant difference test. When the interaction between the fixed effects was significant, the least-square means were compared using Tukey’s method. Results were deemed significant when *p* ≤ 0.05 and trends when 0.05 < *p* ≤ 0.10.

## 3. Results

### 3.1. Maternal Responses

There was a ~35% additional (*p* < 0.01) DMI for CON pregnant beef cows at mid-gestation (day 100 to day 200 of gestation). The average total intake in this period was 6.01 and 7.96 kg (on a DM basis) for RES and CON cows, respectively. In the late gestation (day 201 of gestation to parturition), CON cows had ~18% greater DMI than RES cows (*p* < 0.01; RES = 6.59 and CON = 7.84 kg of DM). The CP (RES = 0.27 and CON = 0.88 kg per day) and TDN (RES = 2.75 and CON = 4.97 kg per day) intakes were higher (*p* < 0.01) for CON cows in the second third of pregnancy. However, in the last third of pregnancy, these parameters were similar in RES and CON (*p* ≥ 0.34). No effects of interactions between maternal protein supplementation and offspring sex were identified on the DM, CP, and TDN intakes in the mid or late gestation (*p* ≥ 0.10). The CON average intake was equivalent to 70% of protein and 50% of energy requirements for RES cows (calculated according to the Nutrient Requirements of Zebu and Crossbred Cattle—BR-CORTE 3.0 [46]) during mid-gestation.

At the beginning of the treatment application, the pregnant beef cows’ BW and BCS were similar between RES and CON groups (*p* ≥ 0.67; Figure 1 and Figure 2). In contrast, at 200 (end of treatment application) and 270 days (pre-parturition) of gestation, the maternal body weights were ~17% and ~14% greater for CON than for RES cows, respectively (*p <* 0.01; Figure 1). Consistently, CON pregnant beef cows also presented greater BCS at 200 and 270 days of gestation compared to the RES group (*p <* 0.01; Figure 2). No PS x OS interactions were verified in the performance parameters of pregnant beef cows (*p* ≥ 0.34).

### 3.2. Offspring Performance

The 120- and 210-day-old CON calves were 12 kg and 17 kg heavier than RES calves (*p* ≤ 0.04), respectively. The average BW at 120 days of age was 126 and 138 kg for RES and CON, and that at 210 days of age was 197 and 214 kg for RES and CON, respectively. In the backgrounding, RES males were ~22 kg lighter than CON males (RES: 255 and CON: 277 kg, *p* < 0.001). Within the female group, RES progenies were ~16 kg lighter than CON progenies (RES: 216 and CON: 232 kg, *p* = 0.05). In the growing 1 phase, CON males remained heavier than RES offspring (RES: 329 and CON: 356 kg, *p* < 0.001), and CON females remained heavier than RES offspring (RES: 271 and CON: 289 kg, *p* < 0.01). In the growing 2 phase, no maternal protein supplementation status effects on the offspring performance data were found (*p* = 0.07). The average BW in this period was 359 and 376 kg for RES and CON offspring, respectively. No PS × OS interaction was detected in the offspring’s body weight (*p* ≥ 0.22). 

### 3.3. Milk Yield and Composition

The CON and RES beef cows had similar milk yields at 120 (*p* = 0.55) and 210 days in milk (DIM, *p* = 0.80). At 120 DIM, RES and CON cows produced 9.04 and 9.53 kg of daily milk. At weaning, the RES and CON daily milk production was 6.70 and 6.54 kg per day, respectively. No PS × OS interaction was detected for milk production (*p* ≥ 0.66). Cows nursing male and female calves also had similar milk yields (*p* ≥ 0.72). No PS or PS × OS effects were found on milk component concentrations at 120 (*p* ≥ 0.44) and 210 (*p* ≥ 0.38) DIM. At 120 DIM, the average milk fat, protein, and lactose percentages were 4.14 and 4.07%, 3.38% and 3.42%, and 4.83% and 4.75% for RES and CON, respectively. At weaning, milk fat, protein, and lactose percentages were 4.74% and 4.84%, 3.45% and 3.49%, and 4.61% and 4.63%, respectively.

### 3.4. Dry Matter Intake and Feed Efficiency

No MN × OS interaction was detected for the feed intake (*p* > 0.05; Table 3). Male calves consumed an additional ~0.6 kg DMI at 120 days of age (*p* = 0.02) compared to females, without discriminated differences for milk, pasture, and supplement intake (*p* > 0.05; Table 3). At weaning, CON offspring tended to have ~11% greater total DMI (*p* = 0.06) and 12% higher pasture intake (*p* = 0.10) compared to the RES group. In the same period, male calves tended to present ~10% and 13% higher total DMI (*p* = 0.06) and pasture intake (*p* = 0.10), respectively. No effects of maternal protein status on dry matter intake were detected during the feedlot period (*p* > 0.05). The 120-day-old male calves presented ~25% greater (*p* = 0.01) DMI/BW than females. At weaning, CON calves tended (*p* = 0.08) to present ~11% additional DMI/BW compared to RES calves. In the growing 1 phase, within female groups, CON females had ~1.5 g of additional feed per kg of BW compared to RES females (Table 3). Maternal protein dietary status only affected the offspring’s feed efficiency for gain at the backgrounding (Table 3). Within female groups, RES offspring were more efficient (RES females: 0.182 vs. CON females: 0.174 g/day of BW per kg of DMI/day, *p* = 0.01). In contrast, within male groups, CON offspring demonstrated greater efficiency for body gain (RES males: 0.210 vs. CON males: 0.214 g/day of BW per kg of DMI/day, *p* = 0.05).

### 3.5. Components of Diet Intake and Apparent Total-Tract Digestibility

The 120-day-old males had an additional intake of ~75 g/day of CP (*p* = 0.05), while at weaning, they presented an additional intake of OM ~595 g/day compared to females (*p* = 0.01; Table 4). At weaning, the maternal protein dietary status tended to affect the OM and affected the CP intake, with additional intake of ~420 and ~100 g/day for CON compared to RES animals (*p* ≤ 0.08; Table 5), respectively. During the digestibility trial performed in the backgrounding, no effects of maternal protein supplementation were found on the DM or nutrient intake (*p* ≤ 0.55). In the growing 1 phase, within the male groups, CON males had ~3% greater OM and CP intake compared to RES males (*p* ≤ 0.05). In the same phase, within the female groups, CON females tended to present 9% additional OM intake (*p* = 0.08) and greater (*p* ≤ 0.04) CP and NDF intake than RES females (Table 5). In the growing 2 phase, males consumed ~655 g/day of OM more than females. Moreover, in the same period, an PS × OS interaction was observed for TDN intake, where RES females and CON males had the lowest daily intake (*p* ≤ 0.001; Figure 3).

Regarding apparent digestibility parameters, at 210 days of age, the CON offspring had ~3.5% greater NDF digestibility (*p* = 0.01). In this period, a tendency of PS × OS interaction was detected (*p* = 0.07) for OM digestibility (Figure 4). In the backgrounding and growing 1 phases, all total-tract digestibility parameters were similar between RES and CON offspring within the male (*p* ≥ 0.20) and female (*p* ≥ 0.15) groups (Figure 4). Interaction between maternal nutrition and offspring sex was verified for digestibility of major nutrients during the growing 2 phase (*p* ≤ 0.02; Figure 4). Overall, lower digestibility coefficients were found for CON males compared to RES males.

### 3.6. Feeding Behavior

The 100-day-old calves from CON cows spent ~8.7 additional minutes per day eating the supplement (*p* = 0.05). A trend of interaction between maternal nutrition and offspring sex (*p* = 0.07) was observed for rumination time in this same period, where RES females spent less time per day in this activity than the CON females.

At weaning, the offspring from the RES group spent ~1:45 additional hours per day in idleness and ~1:10 h less per day in other activities (*p* ≤ 0.02; Table 4). At 360 days of age, within the male groups, CON males spent and tended to spend ~15 and 13 additional minutes per day ruminating (*p* = 0.01) and in other activities (*p* = 0.08) than RES males, respectively. In contrast, in the same period, the RES females spent 35 min more per day in other activities than CON females (Table 4).

## 4. Discussion

The effectiveness of the maternal nutritional treatment application was demonstrated by the responses verified on the maternal body weight and body condition score during pregnancy. Our study demonstrated that maternal protein supplementation during mid-gestation causes changes in the feed intake pattern of the offspring. According to the theory of the thriftiness genotype proposed by Hales and Barker [47], sons of mothers who faced nutritional insults during pregnancy are more efficient in nutrient acquisition and storage. Indeed, studies have indicated that offspring from dams in such conditions may present hyperphagia in later life [15,21,48]. A study using ewes as a model has shown that nutritional restriction of energy and protein during early to mid-gestation resulted in hyperphagic lambs with greater feed efficiency [49]. In contrast with these findings, our responses showed a different pattern from those reported in the current literature, where the highest intake was verified in offspring from well-nourished dams, especially in the cow–calf phase. Moreover, our data interestingly showed that the feed efficiency for gain as a function of maternal protein status during the most accentuated phase of lean mass growth in beef cattle (the period before puberty) happened in different patterns for males and females. In the growing 1 phase, females exposed to nutritional insults during intrauterine development were more efficient in converting the feed intake into animal products, in accordance with the thriftiness genotype theory. In contrast, within male groups, CON offspring presented greater feed efficiency for gain than RES. Therefore, this is further evidence that fetal programming responses are sex-dependent.

Our performance data showed that protein supplementation for pregnant beef cows fed low-quality forage improved the BW of CON offspring over their growth trajectory. This is consistent with previous studies with beef cattle, in which strategic supplementation for pregnant cows improved their offspring growth parameters [6,50]. Mid-gestation (when treatments were applied) is the intrauterine development window in which fetal myogenesis occurs [51]. Furthermore, it is well documented in the scientific literature that feeding strategies such as protein supplementation may act as a key modulator of muscle fiber hyperplasia [8,10], which in turn increases the offspring’s growth potential in post-natal life [11,52]. This could lead to greater nutritional requirements for offspring from well-nourished dams [13], increasing their intake level proportional to their growth potential, which explains the greater DMI for CON offspring.

We have also observed that maternal protein supplementation at mid-gestation affects the nutrient digestibility in the offspring, and these effects were sex-dependent. Interestingly our results demonstrated that the nutrient digestibility of young males from RES dams was reduced during weaning. Moreover, when males from supplemented dams received high-concentrate diets in the growing 2 phase, the digestibility of all nutrients was reduced. This pattern suggested a compensatory mechanism in the digestion and absorption of high-energy diets in RES males in the long term. Consistent with our findings, a recent study has shown that small intestine mass per kg of BW, organ length, villus length, and permeability increase in offspring from dams that were nutritionally restricted during gestation [23,53]. Thus, these reports show a greater absorptive capacity in animals from undernourished dams. Furthermore, Cruz et al. [24] demonstrated that changes promoted by prenatal diet involving the small intestine persist in adult life. These authors showed that males from restricted zebu beef cows without protein supplementation from mid-gestation had a greater intestine length and also a greater expression of key genes related to glucose and fatty acid absorption in the small intestine. Although the expression of key genes related to nutrient absorption was not measured in this study, our observations agree with these responses. Furthermore, the effects of protein supplementation of the dams during mid-gestation on nutrient digestibility in females were more discreet. Therefore, our findings suggest that females seem to be less susceptible to changes in digestibility than males.

When milk production of the dam is not sufficient to meet the demand of the offspring, calves try to compensate for this condition by spending more time grazing [54]. In this study, milk production and composition did not change between RES and CON cows. Thus, this point agrees with the absence of maternal nutrition effects on the suckling and grazing times of calves during the cow–calf phase. At 100 days of age, CON calves spent more time eating supplements than RES calves. Yet, from a practical point of view, this change was small, which corroborates the lack of maternal nutrition effects on supplement intake verified in the digestibility trial performed at 120 days of age. Moreover, at 100 days, there was a trend of interaction between maternal nutrition and offspring sex for ruminating time, showing that females from dams supplemented during mid-gestation spent more time ruminating than females from unsupplemented dams. Nevertheless, these effects on rumination time were not persistent in the long term for females; no effect was detected on the feeding behavior trial performed at 360 days for female groups. However, within the male groups, the CON males fed diets with high concentrate levels (during growing 2) showed higher time in rumination activity than RES males, which is favorable from the production point of view. Consistently, the greater time in idleness for the RES group and the greater time in other activities for the CON group at weaning show that offspring from dams supplemented during mid-gestation exhibit a feeding pattern more compatible with achieving production goals.

Regarding sexual dimorphism’s effects on feeding behavior, this work showed that young calves exhibit sex-specific differences. Although males and female dams did not differ in their daily milk production at the points evaluated, 100-day-old males spent more time per day sucking than females. Similar behavior was verified by Costa et al. [55], who reported that this response occurs due to the higher demand for milk for males than for females once they are heavier. It is possible that the greater time eating supplement verified for males and the greater time in idleness for females at 100 days were also a reflection of the higher growth potential of males.

## 5. Conclusions

Our results demonstrated that fetal programming responses occur in a sex-dependent manner. Protein supplementation for pregnant beef cows fed low-quality forage results in greater offspring performance and carryover effects on their intake pattern. The maternal dietary protein status during mid-gestation reprograms the male offspring’s nutrient use later in life, promoting compensatory responses in their nutrient digestibility to improve tissues’ nutrient access. However, this evolutionary strategy is not enough to promote performance compensation. In summary, this study demonstrates the importance of maternal nutrition during the gestational period.

## Figures and Tables

**Figure 1 animals-12-02865-f001:**
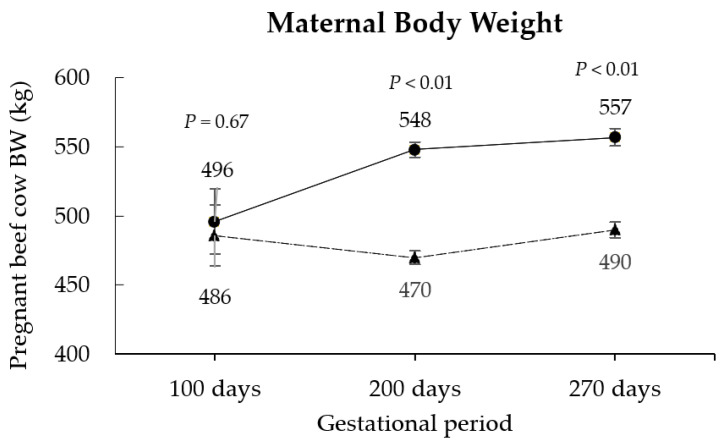
Effects of maternal protein supplementation on pregnant beef cow body weight (kg). RES (unsupplemented cows at mid-gestation, *n* = 24); CON (cows supplemented with protein at mid-gestation, *n* = 19). The symbols represent the average data from CON (●) and RES (▲) groups.

**Figure 2 animals-12-02865-f002:**
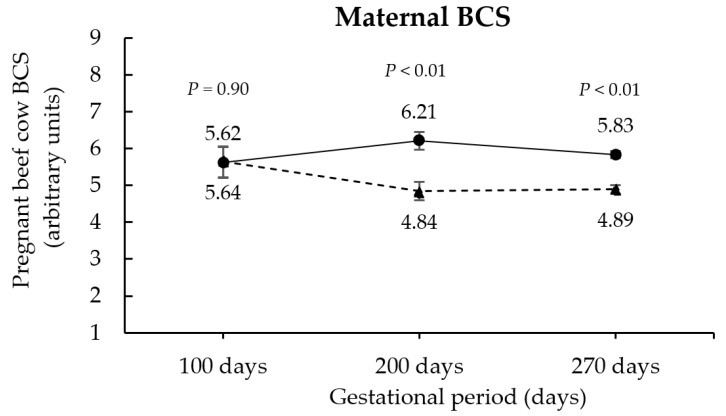
Effects of maternal protein supplementation status on pregnant beef cow body condition score. RES (unsupplemented cows at mid-gestation, *n* = 24); CON (cows supplemented with protein at mid-gestation, *n* = 19). The symbols represent the average data from CON (●) and RES (▲) groups.

**Figure 3 animals-12-02865-f003:**
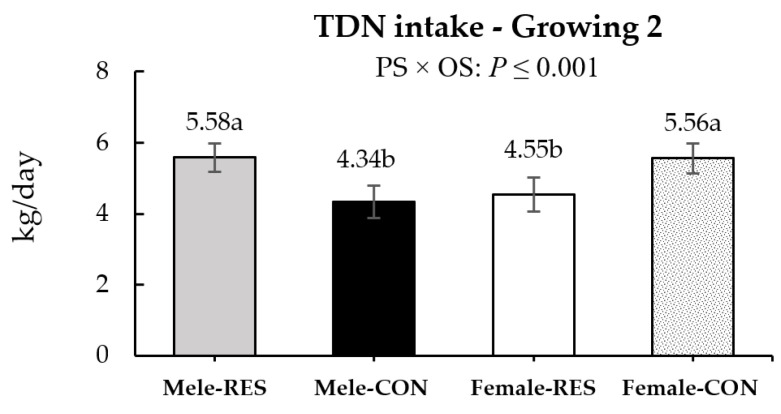
PS × OS interaction effect on the offspring TDN intake in the growing 2 phase. Bars represent mean ± SEM. ^a,b^ Different sub-scripts represents different means (*p* < 0.05).

**Figure 4 animals-12-02865-f004:**
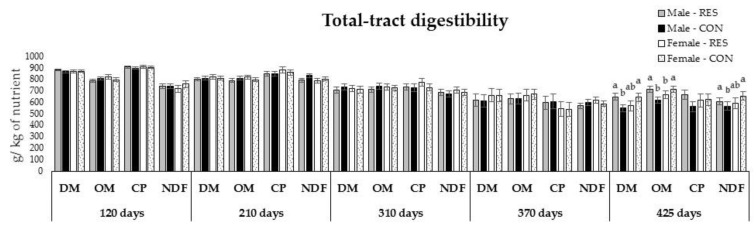
Dry matter (DM), organic matter (OM), crude protein (CP), and neutral detergent fiber (NDF) apparent total-tract digestibility. Bars represent mean ± SEM. ^a,b^ Significant differences between the groups (*p* < 0.05).

**Table 1 animals-12-02865-t001:** The chemical composition of the feedstuffs in both experimental replicates (expressed in the table as the mean ± standard deviation).

	Mid-Gestation (Day 100 to Day 200, Period of Treatment Application)		Late Gestation (Day 100 to Parturition)
Chemical Composition of Feedstuffs, g/kg of DM	Corn silage + Sugarcane Bagasse	Supplement	Corn Silage
Dry matter	418 ± 5.82	881 ± 0.70	330 ± 2.88
Organic matter	951 ± 2.67	958 ± 0.90	941± 1.37
Crude protein	53.3 ± 2.30	400 ± 1.44	72.2 ± 0.42
Ash and protein-free neutral detergent fiber	631 ± 10.60	213 ± 0.18	549 ± 3.62
Non-fibrous carbohydrates	242 ± 5.61	342 ± 2.20	291 ± 2.14
Ether extract	24.1 ± 1.10	41.2 ± 0.32	29.2 ± 0.40

**Table 2 animals-12-02865-t002:** Ingredients and chemical composition of the feedstuffs used in the feedlot.

	Backgrounding ^1^		Growing 1 ^2^		Growing 2 ^3^
	Male	Female	Male	Female	
Ingredients, g/kg of DM					
Corn Silage	717	717	650	650	309
Ground corn	129	152	204	232	577
Soybean meal	116	92.5	106	77.0	101
Urea	6.75	6.75	8.10	8.10	2.88
Ammonium sulfate	0.75	0.75	0.90	0.90	0.32
Mineral nucleus ^4^	30.0	30.0	30.0	30.0	10.0
Chemical composition of experimental diet, g/kg of DM					
Concentrate					
Dry matter	910 ± 10	900 ± 7	900 ± 16	896 ± 19	918 ± 18
Organic matter	829 ± 73	865.9 ± 21	850 ± 18	890 ± 26	930 ± 24
Crude protein	402 ± 107	401 ± 89	315 ± 22	243 ± 31	220 ± 4
Ash and protein-free neutral detergent fiber	144 ± 24	181 ± 82	151 ± 16	231 ± 24	253 ± 27
Non-fibrous carbohydrate	238 ± 173	263 ± 35	363 ± 50	398 ± 39	427 ± 60
Ether extract	16.4 ± 4	21.3 ± 7	20.6 ± 6	17.9 ± 6	30.9 ± 6
Corn Silage					
Dry matter	322 ± 94		347 ± 138		329 ± 92
Organic matter	949 ± 7		883 ± 84		905 ± 80
Crude protein	92.7 ± 14		72.0 ± 18		80.3 ± 20
Ash and protein-free neutral detergent fiber	558 ± 77		547 ± 81		554 ± 77
Non-fibrous carbohydrates	275 ± 79		245 ± 141		249 ± 105
Ether extract	25.4 ± 7		20.0 ± 4		24.9 ± 6

^1^ Backgrounding phase = 255–320 days, ^2^ growing 1 phase = 321–381 days, ^3^ growing 2 = 382–445 days, ^4^ Nutronbeef Maxima Marathon (Cargill Animal Nutrition, Itapira, SP, Brazil). Assurance levels per kilogram of product were: 220 g Ca (max); 200 g Ca (min); 10 mg Co (min); 500 mg Cu (min); 6.60 mg Cr (min); 24 g S (min); 333 mg Fe (min); 18 g P (min); 17 mg I (min); 1500 mg Mn (min); 835 mg monensin; 6.60 mg Se (min); 50 g Na (min); 100,000 UI vitamin A; 13,300 UI vitamin D3; 233 UI vitamin E; 2333 MG Zn (min). Dietary composition in both replicates of the experiment is expressed in the table as the mean ± standard deviation.

**Table 3 animals-12-02865-t003:** Effects of maternal protein supplementation status (PS) and offspring sex (OS) on the intake and DMI/kg of BW during the cow–calf and feedlot phases.

Item	Males		Females		SEM	*p*-Value				
	RES	CON	RES	CON		PS			OS	PS × OS
						Males	Females	General		
Intake, kg of DM/day										
120 days										
Total DMI	2.47	2.19	1.72	1.74	0.35	-	-	0.45	0.02	0.35
Milk intake	1.44	1.21	1.20	1.22	0.11	-	-	0.32	0.25	0.18
Pasture intake	0.91	0.91	0.61	0.74	0.15	-	-	0.61	0.11	0.58
Supplement intake	0.10	0.23	0.20	0.12	0.07	-	-	-	-	0.10
210 days										
Total DMI	4.13	4.65	3.80	4.18	0.45	-	-	0.06	0.10	0.77
Milk intake	0.85	0.71	0.90	0.91	0.15	-	-	0.53	0.24	0.43
Pasture intake	2.88	3.07	2.40	2.86	0.38	-	-	0.10	0.09	0.49
Supplement intake	0.40	0.72	0.42	0.43	0.12	-	-	0.14	0.22	0.19
Backgrounding	5.63	5.50	5.41	5.67	0.77	0.83	0.91	-	-	-
Growing 1	7.25	6.96	7.15	7.60	1.36	0.55	0.12	-	-	-
Growing 2	8.43	8.53	7.86	8.11	0.39	-	-	0.53	0.16	0.79
DMI/BW, g of DM/kg of BW										
120 days	19.9	18.2	14.93	15.5	1.34	-	-	0.59	0.01	0.30
210 days	19.9	21.5	17.8	20.5	2.02	-	-	0.08	0.22	0.64
Backgrounding	22.8	22.5	22.1	22.9	2.99	0.78	0.88	-	-	-
Growing 1	22.8	22.2	23.1	24.6	4.16	0.21	<0.01	-	-	-
Growing 2	23.2	23.8	22.1	22.9	1.04	-	-	0.34	0.24	0.92
Feed efficiency for gain, g/day of BW per kg of DMI/day										
120 days	0.323	0.360	0.388	0.397	0.03	-	-	0.34	0.12	0.55
210 days	0.235	0.239	0.269	0.228	0.02	-	-	0.33	0.22	0.56
Backgrounding	0.210	0.214	0.182	0.174	0.01	0.05	0.01	-	-	-
Growing 1	0.151	0.153	0.141	0.138	0.03	0.21	0.11	-	-	-
Growing 2	0.130	0.134	0.132	0.125	0.02	-	-	0.91	0.77	0.61

Abbreviations: RES (restricted) = offspring from unsupplemented cows; CON (supplemented) = offspring from supplemented cows from 102 ± 5 to 208 ± 6 days of gestation.

**Table 4 animals-12-02865-t004:** Effects of maternal protein supplementation status (PS) and offspring sex (OS) on the calves’ feeding behavior.

Item	Males		Females		SEM	*p*-Value				
							PS		OS	PS × OS
	RES	CON	RES	CON		Males	Females	General		
100 days, min/day										
Suckling	24.5	26.7	15.2	22.6	6.57	-	-	0.20	0.09	0.48
Eating supplement	20.3	29.9	12.8	20.5	4.98	-	-	0.05	0.06	0.83
Grazing	240	234	244	233	28.1	-	-	0.34	0.79	0.85
Ruminating	231	219	196	243	32.6	-	-	0.28	0.75	0.07
Idleness	773	782	862	815	58.3	-	-	0.45	0.03	0.26
Other activities	143	159	107	113	29.6	-	-	0.58	0.04	0.81
210 days, min/day										
Suckling	26.4	23.6	25.3	25.0	4.36	-	-	0.68	0.97	0.74
Eating supplement	37.1	41.6	29.7	27.5	8.05	-	-	0.87	0.13	0.63
Grazing	368	386	400	454	67.0	-	-	0.29	0.16	0.59
Ruminating	239	234	187	238	28.7	-	-	0.21	0.21	0.15
Idleness	698	646	748	591	45.5	-	-	<0.01	0.95	0.13
Other activities	136	171	115	220	57.7	-	-	0.02	0.65	0.21
360 days, min/day										
Eating	198	186	206	199	11.6	0.23	0.94	-	-	-
Ruminating	395	410	342	420	28.9	0.01	0.27	-	-	-
Idleness	739	709	729	728	74.1	0.68	0.77	-	-	-
Other activities	106	119	126	91	78.5	0.80	0.50	-	-	-

Abbreviations: RES (restricted) = offspring from unsupplemented cows; CON (supplemented) = offspring from supplemented cows from 102 ± 5 to 208 ± 6 days of gestation.

**Table 5 animals-12-02865-t005:** Effects of maternal protein supplementation status (PS) and offspring sex (OS) on diet component intake during the digestibility trials, expressed in kg/day.

Item	Males		Females		SEM	*p*-Value				
							PS		OS	PS × OS
	RES	CON	RES	CON		Males	Females	General		
120 days										
OM	1.07	1.24	0.93	1.06	0.18	-	-	0.34	0.42	0.91
CP	0.54	0.53	0.47	0.45	0.03	-	-	0.63	0.05	0.89
NDF	0.70	0.71	0.51	0.59	0.13	-	-	0.68	0.24	0.68
210 days										
OM	2.64	3.04	2.03	2.46	0.24	-	-	0.08	0.01	0.95
CP	0.69	0.78	0.63	0.73	0.05	-	-	0.04	0.18	0.85
NDF	2.32	2.51	1.96	2.32	0.37	-	-	0.14	0.16	0.66
310 days—Backgrounding Phase										
DM	6.08	6.30	6.06	6.11	1.00	0.96	0.67	-	-	-
OM	6.09	5.96	5.94	6.09	0.94	0.97	0.99	-	-	-
CP	1.21	1.12	1.11	1.16	0.30	0.94	0.63	-	-	-
NDF	2.63	2.53	2.66	2.68	0.18	0.55	0.73	-	-	-
TDN	4.24	4.25	4.37	4.22	0.89	0.81	0.93	-	-	-
370 days—Growing 1 Phase										
DM	7.55	7.76	7.40	7.88	1.68	0.43	0.29	-	-	-
OM	7.28	7.52	7.21	7.85	1.62	0.50	0.08	-	-	-
CP	1.19	1.23	1.02	1.04	0.26	<0.01	<0.01	-	-	-
NDF	3.05	3.13	3.15	3.33	0.39	0.91	0.04	-	-	-
TDN	4.49	4.68	4.41	4.91	1.65	0.63	0.24	-	-	-
425 days—Growing 2 Phase										
DM	9.57	9.61	9.06	9.51	0.83	-	-	0.62	0.62	0.65
OM	9.22	9.19	8.65	8.45	0.55	-	-	0.72	0.05	0.79
CP	1.81	1.86	1.70	1.72	0.16	-	-	0.71	0.16	0.83
EE	0.28	0.28	0.29	0.29	0.02	-	-	0.97	0.55	0.88
NDF	3.35	3.44	3.09	3.24	0.28	-	-	0.38	0.12	0.85
TDN	5.58	4.35	4.55	5.56	0.48	-	-	-	-	<0.001

Abbreviations: RES (restricted) = offspring from unsupplemented cows; CON (supplemented) = offspring from supplemented cows from 102 ± 5 to 208 ± 6 days of gestation.

## Data Availability

Data will be made available upon reasonable request to the corresponding author.

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
