# Peer review of "Effects of Maternal Protein Supplementation at Mid-Gestation of Cows on Intake, Digestibility, and Feeding Behavior of the Offspring"

_animals, 2022, doi:10.3390/ani12202865_

Round 1

Reviewer 1 Report

The objective of the work was to evaluate the effect of protein supplementation during mid-gestation and calf sex on intake, behavior, digestibility and feed efficiency of Zebu-beef cows’ offspring and the work was repeated in two years. The topic of research is very interesting, the hypothesis is clearly, and the work is well written in general terms, although sometimes not easy to follow due to the number of results presented.  However, there are some aspects of the work, specifically on M&M, that need to be revised and/or adjusted. 

Major concerns:

1.     Protein supplementation is applied during mid-gestation to beef cows, and although, as presented in the M&M section, diets (thus, treatment) are offered individually, and are described correctly in terms of components and chemical composition, cow average (+/-SE) DM, protein and energy intake should be presented as it is important information related to treatments. We understand that protein and energy intake are presented as % of requirements on lines 110-112 but it could be important to present the actual values and a comparison between CON and SUP groups. In addition, no information about changes in cow BCS and BW during gestation and their comparison between groups is presented. Finally, means of dietary chemical composition (Table 1 and 2) are presented without including a variation measurement. Please include it as it was repeated for 2 years.  

2.     Estimation of calf forage and supplement intake (during the cow-calf phase) based on two external markers to estimate with one forage (TiO2) and with the other one supplement (CrO2) need to be validated. In my knowledge this is not possible.  References indicated on the manuscript refer to the use of only one marker to detect total DMI (lines 165-177). This would impact on the results, as if the methodology presented cannot be sustained, DMI as well as diet components intake and digestibility at 120 and 210 days should be re-analyzed (L309-317; Table 4; feed to gain plots; L342-349; Table 5). 

3.     The measurement of liver size does not provide important information. Please delete. It deviates the focus of the work.  In M&M (lines 234-244) and in the result section (lines 392-396).

4.     Feeding behavior.  Please provide more information: how many observers? 48 h: 2 complete days, daylight and night? How? ) (lines 246-252).  How were scan-observations converted to continuous time? Was probability of occurring different activities analyzed? Was probability of occurring different activities consistent with estimated time spent on different activities? 

5.     Statistical analysis:  In my understanding sex and diets and confounded during backgrounding and growing 1 phases, they cannot be separated by statistical analysis. Please clarify (L261-263). How many data were removed? (lines 276-277). 

6.     Results: 

a.      Lines 309-317, Table 4 and Figure 3: if necessary, modify according with comment 2.

b.     Line 300 and Figure 3: how did the authors separate the diet vs. sex effect during backgrounding and phase 1. See comment 5.  

c.      Figure 2: delete, it does not provide valuable information. 

d.     Lines 342-349), Table 5, Lines 355-357; Table 6: if necessary, modify according with comment 2.

e.     Table 5: Please delete information for 310 and  370 days as no differences were observed due to MN, OS or their interaction. Component intake general mean values can be presented on the text to make this table friendlier. How were TDN values estimated? It could be more interesting to present digestible energy intake. It is surprising the high significance of the interaction MN x OS on this variable at 425 days. Please delete TDN intake of the Table and present a plot with the interaction results. TDN intake combine component intake and digestibility. 

f.      Table 6: digestibility of TDN does not make sense. Delete this Table and keep Figure 4 as information is repeated and Figure 4 summarize better the results. 

g.      Table 7: delete data for females and males when the interaction was <0.10 and present the in a figure if needed or in the text. The table it’s too crowded with information, and these data make it confusing. 

7.     Discussion and conclusion:   these sections must be reviewed again if results are re-analyzed. 

Author Response

Dear Reviewer,

Please, se the attachment.

Kindly regards.

Mateus Pies Gionbelli.

Reviewer 2 Report

General Comments: Based on the below comments, I am recommending reconsideration after major revisions. The authors present interesting data characterizing the influence of maternal protein supplementation on offspring feed intake, digestibility, and feeding behavior; however, major flaws were identified. The description of the experimental design and methods are unclear and should be revised accordingly (see below). Following these revisions, the validity of the methods employed in this study can be evaluated. In general, grammatical and typographical errors should be addressed throughout the manuscript. 

Simple Summary: The Simple Summary contains speculative statements that should be revised to focus on the findings of this study. 

Introduction: The Introduction can be improved by reviewing literature that aligns more specifically with phenotypes evaluated in this study. 

Materials and Methods: 

·     The treatment name “Maternal Nutrition” should be revised to only reflect maternal protein supplementation status.

·    It is unclear which treatments have restricted crude protein, over-supplemented crude protein, or adequate crude protein in the diet. As used, the term “supplemented” is ambiguous. Is crude protein in the CON treatment group restricted or adequate to meet animal requirements? If it is restricted, then referring to this treatment as a control group is inappropriate. The descriptive terms (restricted/control/supplemented/etc.) used for treatments are inconsistent throughout the manuscript. Please revise accordingly and justify the crude protein amounts used for each treatment. 

·     Please describe the source of the sexed embryos. Was sex-sorted semen used to artificially inseminate cows? If this is the case, please clarify these details and whether the sex of the embryos themselves were determined.

·       The differences between the CON and SUP diets are unclear.  Please describe the source of the increased crude protein in the SUP diet. The nutrient analysis should be presented for both the CON and SUP treatment groups. As it is, it appears that other components outside of crude protein may be different between maternal dietary treatments, raising concerns that differences in postnatal phenotypes of calves may be due other differences in maternal nutrition between treatment groups.

·        Was daily feed intake recorded for dams in each treatment?

·      Please describe the pasture allocations at all stages of the study. Were all animals kept in a single pasture or replicated pastures? How many pastures were used? How was treatment represented among pastures at each stage of the study?

·     Section 2.2.4. Liver Ultrasound Scans: Please clarify throughout the manuscript that measures of liver size are estimates.

·      Section 2.2.5. Feeding Behavior: The authors should revise and expand this section substantially to explain how subjective measures of feeding behavior were determined in a consistent manner.  

Results: In general, various figures are missing appropriate axis titles, units of measure, P-values, etc. 

Conclusions: Components of the Conclusions section are vague and overstated. Please elaborate on primary findings and what conclusions can be made from those findings.

Author Response

Dear Reviewer,

Kindly regards,

Mateus Pies Gionbelli.

Reviewer 3 Report

Please see comments given in the reviewed attached file of manuscript.

Author Response

Caro Revisor,

Por favor, verifique o anexo.

Atenciosamente,

Mateus Pies Gionbelli.
